# QUALITY DIVERSITY THROUGH HUMAN FEEDBACK

## ABSTRACT

Reinforcement learning from human feedback (RLHF) has demonstrated potential in qualitative tasks lacking clear objectives. However, its full potential is not fully realized when conceptualized solely as a mechanism to optimize for averaged human preferences, which may not facilitate tasks that demand diverse model responses, such as content generation. Meanwhile, quality diversity (QD) algorithms, which seek diverse and high-quality solutions, rely heavily on manually crafted diversity metrics. This paper introduces Quality Diversity through Human Feedback (QDHF), an approach that integrates human feedback into the QD framework. QDHF facilitates the inference of diversity metrics dynamically from human judgments of similarity among solutions, and thereby enhances the applicability and effectiveness of QD algorithms. Our empirical studies show that QDHF significantly outperforms state-of-the-art methods in automatic diversity discovery and matches the efficacy of using manually crafted metrics on standard QD benchmarks for tasks in robotics and reinforcement learning (RL). Notably, in a latent space illumination task, QDHF substantially increased the diversity of images generated by a diffusion model and was more favorably received in user studies. We conclude by analyzing QDHF's scalability and the quality of its derived diversity metrics, underscoring its potential to improve exploration and diversity in complex, open-ended optimization tasks.

## 1 INTRODUCTION

Foundation models such as large language models (LLMs) and text-to-image generation models in effect compress vast archives of human culture into powerful and flexible tools, serving as a foundation for diverse down-stream applications Brown et al. (2020); Bommasani et al. (2021). Their promise includes helping individuals better meet their diverse goals, such as exploring their creativity in different modalities and coming up with novel solutions. One mechanism to build upon such foundational knowledge is reinforcement learning from human feedback (RLHF) Christiano et al. (2017), which can make models both easier to use (by aligning them to human instructions), and more competent (by improving their capabilities based on human preferences).

RLHF is a relatively new paradigm, and deployments of it often follow a relatively narrow recipe as maximizing a learned reward model of averaged human preferences over model responses. This work aims to broaden that recipe to include optimizing for interesting diversity among responses, which is of practical importance for many creative applications such as generative text-to-image models Rombach et al. (2022), and may have the potential to further improve complex and open-ended learning tasks (through improved exploration), personalization (to serve individual rather than average human preference), and fairness (to embrace diversity and counter-balance algorithmic biases in gender, ethnicity and more).

Diversity encourages exploration, which is essential for finding novel and effective solutions to complex problems. Without diversity, optimization algorithms might converge prematurely to suboptimal solutions, resulting in getting stuck in local optima or producing only a limited set of responses (*i.e.*, mode collapse). The diversity aspect is especially substantial in quality diversity (QD) algorithms (Pugh et al., 2016; Cully et al., 2015; Lehman & Stanley, 2011b; Mouret & Clune, 2015), where diversity metrics are explicitly utilized to encourage and maintain the variation of solutions during optimization.

Our main idea is to derive distinct representations of what humans find interestingly different, and incorporate this procedure in diversity-driven optimization algorithms. We introduce a new concept, Quality Diversity through Human Feedback (QDHF), which empowers QD algorithms with diversity metrics actively learned from human feedback during optimization. To illustrate this concept, we propose an implementation of QDHF capable of formulating arbitrary diversity metrics using latent space projection, and aligning them with human feedback through contrastive learning (Hadsell et al., 2006; Dosovitskiy et al., 2014).

Our work is inspired by the considerable benefits that the learned reward models have unlocked for RLHF. Analogous to reward functions, diversity metrics are often found qualitative, complex, and challenging to exhaustively specify. While existing QD algorithms demonstrate proficiency in addressing complex search and generative tasks due to their inherent explorative capabilities, their reliance on manually crafted diversity metrics restricts their applicability in real-world open-ended tasks. QDHF aims to bridge this gap, allowing QD algorithms to easily adapt to more challenging tasks by actively learning diversity metrics through iterative exploration and human feedback.

In summary, the main contributions of this work are: 1) Introducing QDHF and its implementation leveraging latent projection and contrastive learning. 2) Demonstrating that QDHF mirrors the search capabilities inherent in QD algorithms with manually crafted diversity metrics and outperforms comparable methods that employ automated diversity detection in benchmark robotics QD tasks. 3) Implementing QDHF in the latent space illumination (LSI) task for image generation, demonstrating its capability in producing diverse, high-quality images utilizing human feedback. 4) Providing an analysis on QDHF's sample efficiency and the quality of its learned diversity metrics.

## 2 PRELIMINARIES

In this section, we cover the basic and most relevant aspects of quality diversity (QD) algorithms (Mouret & Clune, 2015; Cully et al., 2015; Pugh et al., 2016; Lehman & Stanley, 2011b), and refer the readers Appendix A for more detailed descriptions.

### 2.1 QUALITY DIVERSITY

QD algorithms effectively explore the search space by maintaining diverse high-quality solutions and using them to drive optimization. Mathematically, given a solution space $\mathcal{X}$, QD considers an objective function $J : \mathcal{X} \to \mathbb{R}$ and $k$ diversity metrics $M_i : \mathcal{X} \to \mathbb{R}, i = 1, 2, \cdots, k$. The diversity metrics jointly form a measurement space $M(\mathcal{X})$, which quantifies the diversity of samples. For each unique measurement in $M(\mathcal{X})$, the global objective $J^*$ of QD is to find a solution $x \in \mathcal{X}$ that has a maximum $J(x)$ among all solutions with the same diversity measurement.

Considering that the measurement space $M(\mathcal{X})$ is usually continuous, a QD algorithm will ultimately need to find and store an infinite number of solutions corresponding to each location in the measurement space. One common way to mitigate this is to discretize $M(\mathcal{X})$ into an archive of $s$ cells $\{C_1, C_2, \cdots, C_s\}$, which was introduced in MAP-Elites (Mouret & Clune, 2015; Cully et al., 2015) and has been widely adopted in its variants. The QD objective is thus approximated by relaxing the global objective to finding a set of solutions $\{x_i\}, i \in \{1, ..., s\}$, and each $x_i$ is the best solution for one unique cell $C_i$. This (approximated) $J^*$ can be thus formulated as

$$J^* = \sum_{i=1}^{s} \max_{x \in \mathcal{X}, M(x) \in C_i} J(x) \tag{1}$$

### 2.2 QUALITY DIVERSITY WITHOUT PREDEFINED DIVERSITY METRICS

Diversity maintenance in QD algorithms usually relies on manually designed diversity metrics to ensure a varied solution archive. However, such a requirement restricts the applicability of QD in more complex and open-ended domains, where the notion of diversity is likely to be abstract and qualitative. To address this, the concept of QD with automatic discovery of diversity has become popular in recent studies. Instead of using pre-defined diversity metrics, Cully (2019); Meyerson et al. (2016); Grillotti & Cully (2022) proposed AURORA, which utilizes unsupervised dimension reduction techniques to learn diversity metrics directly from the data.

However, one problem with these unsupervised methods is that the derived diversity metrics tend to capture the overall variance in the *existing* data, which may not align well with the diversity required for obtaining novel and superior solutions, making these methods uncapable in solving complex problems. As inspired by recent work in RLHF (Christiano et al., 2017) where reward models learned from human feedback are used to solve abstract and complicated optimization problems, we raise two central research questions: Can human feedback be employed to derive appropriate diversity metrics for QD that benefit optimization? And how does this approach compare to unsupervised or manually crafted metrics in terms of diversity maintenance and effectiveness of search?

To address these questions, we introduce a new paradigm of quality diversity through human feedback (QDHF), where the diversity metrics are learned from human judgment on similarity of the data. Using human feedback is not only more flexible than manually designing a diversity metric, but it also caters to more abstract and complex domains where defining a numeric metric of diversity is challenging. In general, QDHF endeavors to optimize for diverse solutions by leveraging diversity metrics that align with human intuition. In the following section, we introduce a specific way of implementing QDHF through latent space projection and contrastive learning.

## 3   METHODS

In this section, we first introduce our formulation of arbitrary diversity metrics from a general modeling perspective using latent space projection, then describe how to align such metrics with human feedback through contrastive learning. Finally, we propose a quality diversity through human feedback (QDHF) framework that leverages human judgment for quality diversity algorithms as a semi-supervised optimization method.

### 3.1   DIVERSITY CHARACTERIZATION THROUGH LATENT PROJECTION

Recent work (Cully, 2019; Meyerson et al., 2016; Grillotti & Cully, 2022) has shown that unsupervised dimensionality reduction algorithms such as principal component analysis (PCA) and auto-encoder (AE) can be utilized to automatically learn robot behavioral descriptors based on raw sensory data. In our framework, we first introduce a general concept of *diversity characterization*. Given data that contains information of diversity, we implement a latent projection, transforming it into a semantically meaningful latent space. More specifically, given an input vector $x$, we first (optionally) employ a feature extractor $f : \mathcal{X} \to \mathcal{Y}$ where $\mathcal{X}$ denotes the input space and $\mathcal{Y}$ the feature space. Post extraction, a projection function parameterized with $\theta$, denoted as $D_r(y, \theta) : \mathcal{Y} \to \mathcal{Z}$, is applied to project the feature vector $y$ into a more compact latent representation:

$$z = D_r(y, \theta), \text{where } y = f(x). \tag{2}$$

In this context, $\mathcal{Z}$ represents the latent space, wherein each axis or dimension corresponds to a diversity metric derived from the data. The magnitude and direction along these axes capture nuanced characteristics of the original data, offering a compact yet informative representation of $x$. For example, on a single axis, smaller and larger values typically indicate variations in certain diversity characteristics. In this work, we apply linear projection for dimensionality reduction, where the parameters are learned through a contrastive learning process described in the following subsection.

### 3.2   ALIGNING DIVERSITY METRICS WITH HUMAN PREFERENCES

While classic unsupervised dimensionality reduction methods capture significant data variances, the resulting latent representations may not consistently offer meaningful semantics for optimization. Recognizing this shortcoming, QDHF effectively aligns the diversity latent space with human notions of diversity by adopting contrastive learning Chopra et al. (2005); Schroff et al. (2015).

**Contrastive learning.**   Recent work has demonstrated success of using contrastive learning in modeling human preferences for RLHF (Christiano et al., 2017) and image perceptual similarity (Fu et al., 2023). Our framework takes a similar approach. Given a triplet of latent embeddings $z_1$, $z_2$, and $z_3$ and suppose that the human input indicates that $z_1$ is more similar to $z_2$ rather than $z_3$, our intention is to optimize the spatial relations of $z_1$ and $z_2$ in the latent space relative to $z_3$. We use a

triplet loss mechanism, *i.e.*, to minimize the distance between $z_1$ and $z_2$ while maximize between $z_1$ and $z_3$ via a hinge loss. This objective can be formalized as:

$$\mathcal{L}(z_1, z_2, z_3) = \max(0, m + D(z_1, z_2) - D(z_1, z_3)) \tag{3}$$

where $D(\cdot, \cdot)$ represents a distance metric in the embedding space, and $m$ acts as a predetermined margin. Through this approach, the learned latent projections are aligned with both the inherent structure of the data and human notions of similarity.

**Human judgement on similarity.** To accommodate our design of contrastive learning, we leverage the Two Alternative Forced Choice (2AFC) mechanism (Zhang et al., 2018) to obtain judgments on input triplets. When presented with a triplet $\{x_1, x_2, x_3\}$, an evaluator is prompted to discern whether $x_2$ or $x_3$ is more similar to the reference, $x_1$. Importantly, this mechanism works with not only human judgment, but also judgment produced by heuristics and other AI systems, which means that our framework remains universally applicable across varying feedback modalities.

### 3.3 QUALITY DIVERSITY THROUGH HUMAN FEEDBACK

Based on the QDHF concept introduced in Sec. 2.2, we propose an implementation of QDHF through latent space projection and contrastive learning with human judgments.

**Objective.** In QDHF, the diversity metrics $M_{\text{hf},i} : \mathcal{X} \to \mathbb{R}$ are derived from human feedback on the similarity of solutions. Given a solution $x \in \mathcal{X}$, we define $M_{\text{hf},i}(x) = z$ where $z \in \mathcal{Z}$ is the latent representation defined in Eq. 2. The latent space $\mathcal{Z}$ is used as the measurement space $M_{\text{hf}}(\mathcal{X})$, where each dimension $i$ in $\mathcal{Z}$ corresponds to a diversity metric $M_{\text{hf},i}$. We can now specialize Eq. 1 for the objective of QDHF, $J_{\text{hf}}^*$, which is formulated as:

$$J_{\text{hf}}^* = \sum_{i=1}^{s} \max_{x \in \mathcal{X}, z \in C_i} J(x) \tag{4}$$

where $z = D_r(f(x), \theta)$ (Eq. 2). To effectively learn the parameters $\theta$ in latent projection, we utilize the contrastive learning mechanism (Eq. 3).

**Training strategy.** We design two training strategies for both offline learning and online learning scenarios, QDHF-offline and QDHF-online. For QDHF-offline, we assume the human judgment data is collected, and the latent projection is trained before running the QD algorithm. On the other side, QDHF-online uses an active learning strategy, in which case the latent projection is initially trained with randomly sampled human judgment data, but also fine-tuned during the QD process. As QD continuously adds solutions to the archive, we sample random triplets of solutions from the current archive and seek human judgment on the similarity. The online data is used to iteratively fine-tune the latent projection, followed by creating a new QD archive initialized with current solutions but with new diversity descriptors. The frequency of fine-tuning decreases exponentially over time as the learned metrics become more robust. In our experiments, the latent projection is updated 4 times at iteration 1, $10\% \cdot n$, $25\% \cdot$, and $50\% \cdot$ for a total of $n$ iterations. To fairly compare with QDHF-offline, each update consumes 1/4 of the total budget of human feedback.

## 4 EXPERIMENTS

### 4.1 TASKS

We describe our experimental setup across three benchmark tasks in the scope of robotics, RL, and computer vision. For all experiments, we use MAP-Elites (Mouret & Clune, 2015) as the basic QD algorithm. More specific implementation details can be found in Appendix B.

**Robotic arm.** We use the robotic arm domain derived from (Cully et al., 2015; Vassiliades & Mouret, 2018). The primary goal is to identify an inverse kinematics solution for each accessible position of the endpoint of a planar robotic arm equipped with revolute joints. The objective function is to minimize the variance of the joint angles. Correspondingly, a common way of measuring diversity is to record the endpoint's positions in a 2D space. These metrics are computed using the forward kinematics of the planar arm, as detailed in Murray et al. (2017).

**Maze navigation.** Kheperax (Grillotti & Cully, 2023) is a recently developed benchmark environment for conventional QD and QD-RL algorithms, which features a maze navigation task originally proposed in Mouret & Doncieux (2012). The goal of the task is to discover a collection of neural network policy controllers that facilitate the navigation of a robot across varying positions in a maze. A Khepera-like robot is used as the agent in this environment, which is equipped with laser sensors positioned at the robot's facing directions for computing the distance. The maze is designed to be deceptive with immovable walls, making the navigation a challenging optimization task.

**Latent space illumination.** The latent space illumination (LSI) task (Fontaine et al., 2021) is designed for applying QD algorithms to explore the latent space of a generative model. The initial LSI in Fontaine et al. (2021) demonstrates using QD to generate gameplay levels. Fontaine & Nikolaidis (2021) extends LSI to generate images of human faces that match specific text prompts.

In this work, we introduce a new LSI pipeline as a more general black-box optimization task for text-to-image generation. We use Stable Diffusion (Rombach et al., 2022), a latent diffusion model with state-of-the-art capability to generate detailed images conditioned on text descriptions. In this task, the goal is to find high-quality and diverse images by optimizing the initial latent vectors, where the diffusion model is treated as a black-box. The objective is to match a text prompt scored by CLIP, but there is no pre-defined diversity metrics, which makes this task a more challenging and open-ended optimization problem in a practical, real-world setting.

## 4.2 EXPERIMENTAL DESIGN AND EVALUATION

In our experiments, we design two specific scenarios. One where the ground truth diversity metric is available and one where a ground truth diversity metric is not available.

**Tasks with ground truth diversity.** The first scenario leverages a predefined ground truth diversity metric to simulate human feedback, which is derived from the differences between solutions as measured by this metric. The primary reason for using simulated feedback is to facilitate the evaluation of QDHF and enable its comparison with other methods under consistent ground truth diversity metrics. Otherwise, measuring the diversity of solutions consistently across different methods would be challenging. This approach is applied to the robotic arm and maze navigation task, where the ground truth diversity metrics correspond to the position (x and y values) of the arm or robot in a 2D space. The "human judgment" is determined by the similarity of the positions, calculated as the L2 distance from the ground truth measurements.

To validate the effectiveness of our method, we benchmark against AURORA (Grillotti & Cully, 2022; Cully, 2019), which is an unsupervised diversity discovery method designed for QD, and standard QD using ground truth diversity metrics, which offers an oracle control representing the best possible performance of QD. We use four variants of AURORA, encompassing two dimension-reduction techniques: PCA and AE, and two training strategies: pre-trained and incremental. For evaluation, solutions of each method are additionally stored in a separate archive corresponding to the ground truth diversity metrics, which is used exclusively for evaluation. We report QD score (Pugh et al., 2015) (sum of objective values, Eq. 1) and coverage (ratio of filled cells to total cells). The QD score is normalized by the archive size to a 0-100 scale for clarity.

The evaluation is conducted from two settings: 1) solutions contained within the final archive, and 2) solutions discovered by the algorithm throughout its entire search process. Specifically, the first setting evaluates the alignment of learned diversity metrics with the ground truth metrics. The second setting provides insights into the overall efficacy of the search process regardless of how the diversity is measured and maintained. This distinction is essential given QD's fundamental premise of leveraging diversity to enhance optimization.

**Tasks without ground truth diversity.** In the second scenario, there is no ground truth diversity metric and real human feedback data is used, which applies to the LSI task. To facilitate the scalability of our method, instead of using human labelers in the loop, we alternatively utilize the real human feedback data to train a preference model, and use the preference model to source feedback for training and evaluation. Similar approaches have been widely used in recent RLHF applications such as LLM training (Stiennon et al., 2020; Ouyang et al., 2022), where a preference model is trained on human feedback data, and used to predict rewards for fine-tuning the LLM.

Table 1: Results for robotic arm. We report the QD score and coverage for both Archive Solutions (solutions in the final archive) and All Solutions (solutions found throughout training). QDHF-online significantly outperforms all other methods in both QD score and coverage, and closely approaches the search capability of QD with ground-truth diversity when considering all solutions.

| Methods | Archive Solutions | | All Solutions | |
|---|---|---|---|---|
| | QD score | Coverage | QD score | Coverage |
| AURORA-AE (Pre-trained) | $14.3 \pm 6.3$ | $20.3 \pm 6.4$ | $38.5 \pm 12.7$ | $56.2 \pm 6.8$ |
| AURORA-AE (Incremental) | $17.6 \pm 4.1$ | $18.9 \pm 4.1$ | $53.0 \pm 9.4$ | $63.2 \pm 6.2$ |
| AURORA-PCA (Pre-trained) | $14.2 \pm 4.4$ | $19.3 \pm 5.5$ | $38.4 \pm 13.6$ | $54.1 \pm 9.0$ |
| AURORA-PCA (Incremental) | $18.3 \pm 3.4$ | $19.0 \pm 3.5$ | $45.9 \pm 6.4$ | $59.0 \pm 3.7$ |
| QDHF-Offline | $31.8 \pm 4.5$ | $34.1 \pm 4.2$ | $54.5 \pm 4.3$ | $62.7 \pm 2.7$ |
| QDHF-Online | $\mathbf{56.4 \pm 0.9}$ | $\mathbf{59.9 \pm 0.9}$ | $\mathbf{72.5 \pm 0.9}$ | $\mathbf{77.3 \pm 1.2}$ |
| QD (Ground-truth diversity) | $74.8 \pm 0.2$ | $79.5 \pm 0.3$ | $74.8 \pm 0.2$ | $79.5 \pm 0.3$ |

We use the DreamSim as the preference model on image similarity, which is trained on human judgment data from the NIGHTS dataset, both from Fu et al. (2023). For comparisons, we implement a best-of-n approach as the baseline, which generates in total 2,000 images (the same number as those explored by QDHF) randomly from uniform $\mathcal{U}(0, 1)$, and select a solution set of 400 images (the same number of the solution archive in QDHF) with the top CLIP scores. We also propose a heuristic as a stronger baseline that also considers diversity, which works as follows: Starting from the second step, we sample 100 latents randomly, and choose the one with a maximal l2 distance to the previous sample. This approach increases the diversity of the latents.

We conducted LSI experiments using 6 common text prompts to show generality. We provide both quantitative and qualitative results for a comprehensive evaluation. Quantitatively, we report the average CLIP score for assessing the objective, *i.e.*, how well the generated images match the prompt. We also use DreamSim to calculate the pairwise distance between images in the solution to measure diversity. The mean pairwise distance indicates the average separation of images, and the standard deviation indicates the variability in how the images are distributed. For qualitative results, we conduct a user study ($n = 43$) to assess their opinion on 1) which result is more preferred, and 2) which result is more diverse. In addition, we display samples of solutions for qualitative assessment.

### 4.3 RESULTS

**Robotic arm.** For the robotic arm task, detailed results are presented in Table 1. The statistics are accumulated over 20 repeated trials. QDHF-online significantly surpasses AURORA in terms of both QD score and coverage. The results highlight QDHF's capability to enhance QD algorithms over unsupervised diversity discovery methods. Notably, in terms of all solutions, QDHF-online closely matches the performance of QD with a ground truth metric, suggesting QDHF as a potent alternative for QD when manually designed metrics are not available for optimization.

**Maze navigation.** Results for the maze navigation task are shown in Table 2. The statistics are accumulated over 10 repeated trials because maze navigation is more computationally expensive. Similar to robotic arm, we observe that QDHF-online is the best for most metrics, surpassing AURORA and closely matching the performance of standard QD. The results further support the validity of QDHF as an alternative to standard QD with improved flexibility and comparable performance.

**Latent space illumination.** Results for the LSI task are shown in Table 3. The results are summarized over 6 different text prompts, and we include more details in Appendix C. Baseline is the best-of-n approach, and Baseline+ is the best-of-n with enhanced diversity in sampling (Sec. 4.2). Quantitatively, QDHF has a similar CLIP score to both baseline methods, but much higher mean and standard deviation of pairwise distance, which shows that QDHF is able to generate more diverse solutions while maintain the high-quality. We also observe that the Baseline+ heuristic does not produce better diversity, which indicates that there exists strong inductive bias in the diffusion model, and generating diverse images is a challenging latent optimization problem.

Table 2: Results for maze navigation. QDHF also shows superior performance compared to other methods, and a resemblance of the search capability of QD with ground-truth diversity.

| Methods | Archive Solutions | | All Solutions | |
|---|---|---|---|---|
| | QD score | Coverage | QD score | Coverage |
| AURORA-AE (Pre-trained) | $22.3 \pm 0.6$ | $23.0 \pm 0.8$ | $35.9 \pm 0.6$ | $38.1 \pm 0.9$ |
| AURORA-AE (Incremental) | $19.4 \pm 1.2$ | $22.8 \pm 2.0$ | $40.0 \pm 2.1$ | $46.7 \pm 4.9$ |
| AURORA-PCA (Pre-trained) | $22.9 \pm 0.6$ | $23.7 \pm 0.7$ | $35.5 \pm 0.4$ | $37.8 \pm 0.4$ |
| AURORA-PCA (Incremental) | $18.0 \pm 0.7$ | $21.0 \pm 1.1$ | $39.0 \pm 0.8$ | $45.3 \pm 3.5$ |
| QDHF-Offline | $\mathbf{23.7 \pm 0.9}$ | $24.4 \pm 1.1$ | $35.6 \pm 0.6$ | $37.9 \pm 1.1$ |
| QDHF-Online | $22.5 \pm 1.3$ | $\mathbf{27.2 \pm 3.0}$ | $\mathbf{42.0 \pm 1.7}$ | $\mathbf{51.3 \pm 5.5}$ |
| QD (Ground-truth diversity) | $42.7 \pm 2.7$ | $52.6 \pm 7.0$ | $42.7 \pm 2.7$ | $52.7 \pm 7.1$ |

Table 3: Results for LSI. We report the CLIP score and DreamSim pairwise distance (mean and std.) as quantitative metrics. QDHF demonstrates comparable quality to both baseline methods measured by CLIP score, and significantly better diversity measured by pariwise distance. In the user study, QDHF also outperforms baseline with a considerable margin. (*The qualitative indicators are ratios, and participants had the option to select 'hard to decide' in their responses.)

| Method | Quantitative Metrics | | | Qualitative Indicators* | |
|---|---|---|---|---|---|
| | CLIP Score | Mean PD | Std. PD | User Preference | Perceived Diversity |
| Baseline | 68.85 | 0.420 | 0.106 | 26.7% | 8.91% |
| Baseline+ | 68.90 | 0.419 | 0.105 | N/A | N/A |
| QDHF-online | **69.08** | **0.527** | **0.151** | **54.7%** | **75.2%** |

We also compare QDHF and Baseline qualitatively through a user study with 43 participants, where QDHF also outperforms baseline with a considerable margin on both the user preference ratio and user diversity perception ratio. Notably, we find that most users think QDHF generates more diverse images, and a majority of them also prefers to have such diversity in the solution. An example of the LSI results is depicted in Fig 1. We can observe that images generated by QDHF have more variations, and show visible trends of diversity.

## 4.4 ANALYSIS AND DISCUSSION

**Scalability of QDHF.** In the context of the scalability, determining the necessary sample size of human feedback for desired performance is critical. To investigate this, we first perform an ablation study with varying sample sizes of simulated human judgments on the robotic arm task, and the results are shown in Fig. 2. The left plot demonstrates the relationship between QD score and sample size. It is evident that there is a strong correlation between the QD score and sample size. Diving deeper into this relationship, we evaluate how the latent projection captures diversity and mirrors human judgment. We calculate the accuracy of using the latents to predict the human judgment on a validation set. The right plot reveals a strong correlation between validation accuracy and the QD score, which suggests that the performance of QDHF heavily relies on the efficacy of learning the latent projection. In other words, by evaluating the judgment prediction accuracy during training, we can estimate whether the current labels are enough for QDHF to perform well.

Secondly, we note that the judgment data for QDHF does not need to come entirely from humans. In the LSI experiments, we use a preference model (DreamSim) to source human feedback during training QDHF, and we show that an accurate preference model can be used for QDHF as an alternative to human labelers. Combining these two observations, we conclude that QDHF has good scalability towards more complex tasks because 1) we can anticipate its performance through online validation, and 2) the samples used by QDHF can be sourced from a preference model trained on a fixed amount of human labor.

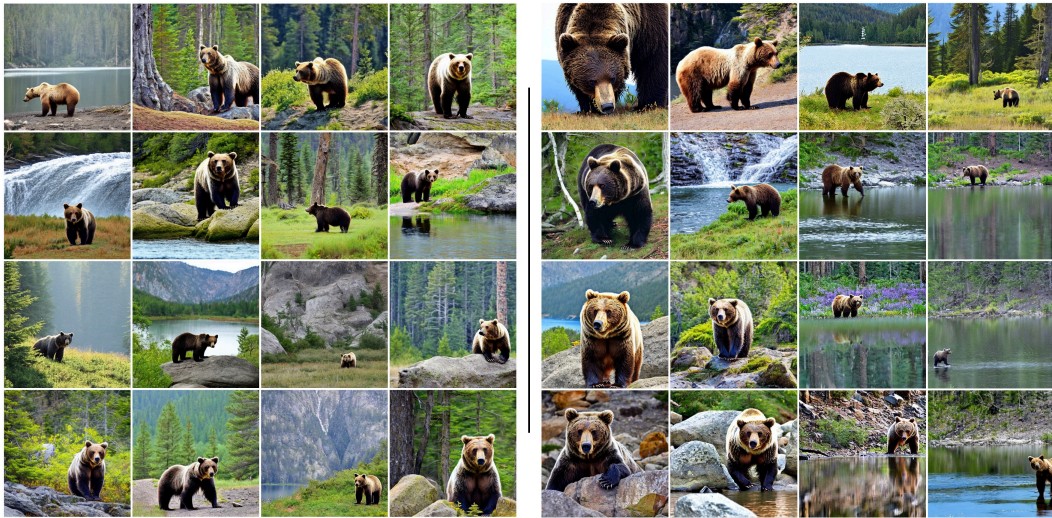

Figure 1: Qualitative result for LSI. The objective prompt is "an image of a bear in a national park". The left 4x4 grid displays images with the highest CLIP scores from randomly generated images. The right grid displays a uniformly-sampled subset of QDHF-online solutions. Qualitatively, images generated by QDHF have more variations, and show visible trends of diversity such as object sizes (large to small along x-axis) and landscape types (rocky to verdant along y-axis, terrestrial to aquatic along x-axis). This selected example closely aligns with the average user preference ratio observed in our user experience study, where QDHF results are about twice as preferred as the baseline.

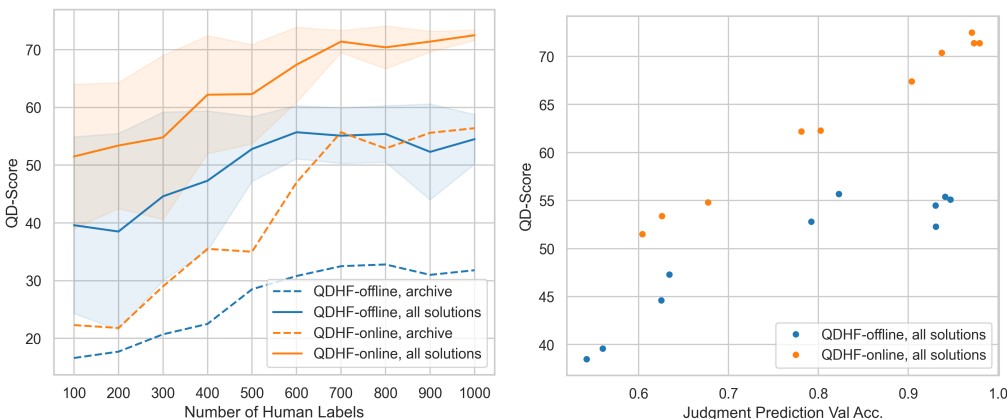

Figure 2: Analysis of varying human feedback sample sizes on robotic arm. "Judgement prediction val acc" is the accuracy of the latent projection in predicting human preferences based on a validation set. There is a direct correlation between QD score and sample size, with QDHF's performance closely tied to the accuracy of latent projection in reflecting human judgment.

**Alignment between learned and ground truth diversity metrics.** We evaluate the alignment of diversity metrics derived by QDHF with the underlying ground truth diversity metrics. In Fig. 3, the archives of QDHF-online and AURORA-PCA (incremental) are visualized for the maze navigation task. Both AURORA and QDHF appear to effectively learn a diversity space reflective of the ground truth. However, QDHF exhibits enhanced capability in discerning the relative distances between solutions, especially in under-explored areas. This suggests that while AURORA and QDHF both exhibit robust exploration capabilities within their learned diversity spaces, QDHF consistently identifies more diverse solutions. This efficacy stems from QDHF's ability to better align its learned diversity space with the ground truth diversity, especially concerning the scales on each axis.

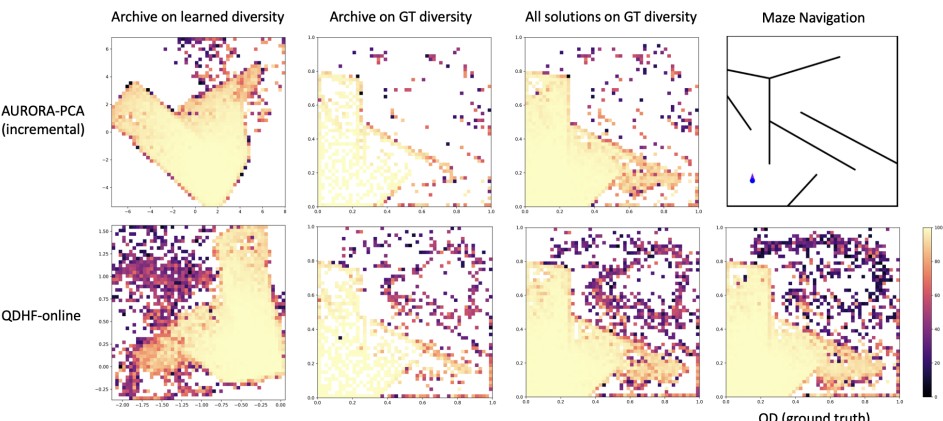

Figure 3: Visualization of the solution archives on different diversity spaces for maze navigation. "GT" stands for ground truth. Each point on the heatmap is a solution with its objective value visualized in color. QDHF fills up the archives with more solutions than AURORA. While both AURORA and QDHF learned a rotated version of the maze as diversity (first column), QDHF is able to more accurately learn the scale of the maze especially in the under-explored area.

## 5 RELATED WORK

**Learning from human feedback.** This work expands upon recent developments in methodologies for aligning models with human objectives. RLHF (Christiano et al., 2017; Ibarz et al., 2018) was initially proposed for training RL agents in simulated environments such as Atari games. It has been later applied to fine-tune or perform one-shot learning on language models for tasks such as text summarization (Ziegler et al., 2019; Stiennon et al., 2020; Wu et al., 2021), dialogue (Jaques et al., 2019), and question-answering (Nakano et al., 2021; Bai et al., 2022; Ouyang et al., 2022), as well as vision tasks such as perceptual similarity measure (Fu et al., 2023). While past work focuses on learning a reward or preference model from human intentions, we propose to learn diversity metrics and use them to drive the optimization process in QD algorithms.

**Quality diversity.** Instead of optimizing for one optimal solution, diversity-driven optimization methods such as Novelty Search (Lehman & Stanley, 2011a) and Quality Diversity (Lehman & Stanley, 2011b; Cully et al., 2015; Mouret & Clune, 2015; Pugh et al., 2016) aim to identify a variety of (top-performing) solutions that exhibit novelty or diversity. Prior work expands on QD by enhancing diversity maintenance (Fontaine et al., 2019; Smith et al., 2016; Vassiliades et al., 2017), search process (Fontaine et al., 2020; Vassiliades & Mouret, 2018; Nordmoen et al., 2018; Sfikas et al., 2021), and optimization mechanism (Kent & Branke, 2020; Conti et al., 2018; Fontaine & Nikolaidis, 2021). This work focuses on problems where, unlike the general assumption in these methods, there is no predefined diversity metrics. Recent work (Grillotti & Cully, 2022; Cully, 2019) explores unsupervised methods for diversity discovery. Our work differs by leveraging human feedback, which is aligned with human interest and often more beneficial for optimization.

## 6 CONCLUSION

This paper introduced Quality Diversity through Human Feedback (QDHF), which expands the reach of QD algorithms through leveraging human feedback to infer measures of diversity. Our empirical results show that QDHF outperforms current unsupervised diversity discovery methods for QD, and compares well to standard QD that utilizes manually-crafted diversity metrics. In particular, applying QDHF in a latent space illumination task shows that QDHF substantially enhances the diversity of images in the text-to-image generation task. Furthermore, we provide an analysis of QDHF's scalability and of the quality of the diversity metrics it learns. For future work, we aim to apply QDHF in more challenging robotics, RL, and generative tasks, with the hope of demonstrating its scalability and effectiveness in complex, open-ended environments.

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
