# OpenReview forum: "Quality Diversity through Human Feedback"
_ICLR.cc/2024/Conference — Submitted to ICLR 2024_

### Official Review · Reviewer_FS1c · 2023-10-13

**Soundness:** 2 fair
**Presentation:** 2 fair
**Contribution:** 3 good
**Rating:** 6
**Confidence:** 3

**Summary:**

Inspired by the success of RLHF to align large language models, the paper introduces Quality Diversity through Human Feedback (QDHF).
The main motivation is to learn a diversity model instead of using adhoc diversity metrics, by analogy with reward modeling in place of hand-engineered rewards in RLHF.
This paper demonstrates empirically that using those diversity models (trained by contrastive learning) improves the results obtained with standard diversity metricsfor automated diversity discovery. Furthermore, when used in image generation tasks, QDHF can generate diverse images for a given caption ("a photo of an astronaut riding a horse on mars").

**Strengths:**

- Novelty: Using diversity modeling from feedback is innovative, and presents a fresh way to look at diversity.
- Results: The study supports empirical evidence indicating that QDHF works.
- Applications: This method can be applied in diverse domains like image generation.

**Weaknesses:**

- The only large scale experiment, on image generation, is actually restricted to a singular prompt with 16 generated images. More experiments are required.
- The papers include "human feedback" in its title, but actually all the experiments are with automatic feedback.
- The paper could benefit from a clearer exposition of implementation choices. The absence of supplementary materials exacerbates this opacity.
- Minors
    * I find the paper quite complex to follow for the readers not well-versed with Quality-diversity.
    * An ablation study highlighting the impact of various design choices would have added depth to the research.

**Questions:**

- Regarding the image generation experiment, I'm unclear about the source of randomness. Is the randomness exclusively attributed to the diffusion process? Or is there also an element of prompt modification? Additionally, could you elucidate how the cells are defined in image generation?

---

> ### Author Response · Authors · 2023-11-22
> **Rebuttal from Authors of Submission6896 to Reviewer FS1c (1/2)**
>
> >**Q1.** The only large scale experiment, on image generation, is actually restricted to a singular prompt with 16 generated images. More experiments are required.
>
> **A1.** We appreciate the feedback. As suggested by the reviewer, we ran the LSI experiment on 5 additional prompts as proof of generality. We also would like to clarify that, the LSI task actually generates a total of 2000 images for each prompt. The qualitative results display samples of top solutions. For the baseline best-of-n method, we select the images with the best CLIP score and display the best 8 (or 16) images in the 2x4 (or 4x4) cells. For our method, we display the images in 4x4 cells by sampling from a 20x20 QD archive, i.e., each displayed image is the one with the best CLIP score in a 5x5 subset of the archive.
>
>
> >**Q2.** The papers include "human feedback" in its title, but actually all the experiments are with automatic feedback.
>
> **A2.** We appreciate the thoughtful comment and have added the following discussion to clarify ways of using human feedback. First, in the robotic arm and maze navigation tasks, the human feedback is simulated from a ground truth diversity metric, which is intended as to offer an oracle control representing the best possible performance of QD, so that we can compare against it. Results show that QDHF could mirror the capability of QD on those standard QD benchmarks.
>
> Secondly, the judgment data for QDHF does not need to be entirely from humans, which offers the scalability of our method toward large-scale tasks. In the LSI experiments, we use a preference model (DreamSim) trained on human feedback data to mimic human feedback during training QDHF, and we show that an accurate preference model can also be used for QDHF as an alternative to online human labelers, which could be very costly. Similar approaches have been widely used in RLHF applications such as LLM training [1,2], where a preference model is trained offline on human feedback data, and used to predict rewards for fine-tuning the LLM.
>
> [1] Stiennon, N., et al., (2020). Learning to summarize from human feedback.
>
> [2] Ouyang, L., et al., (2022). Training language models to follow instructions with human feedback.
>
>
>
> >**Q3.** The paper could benefit from a clearer exposition of implementation choices. The absence of supplementary materials exacerbates this opacity.
>
> **A3.** We appreciate the feedback. We would like to clarify that we did include supplementary materials during submission, which is a .zip file containing an appendix including additional implementation details and the source code to reproduce our results, which should be visible to the reviewers. We hope the reviewer can double-check to see if they have access to the supplementary mateirals, and contact the AC immediately if the issue persists. We have also revised the supplementary material to include more implementation details in the Appendix.

---

> ### Author Response · Authors · 2023-11-22
> **Rebuttal from Authors of Submission6896 to Reviewer FS1c (2/2)**
>
> >**Q4.** I find the paper quite complex to follow for the readers not well-versed with Quality-diversity.
>
> **A4.** We appreciate the valuable feedback. We have revised Sec. 2 and 3 to describe our method from a more general ML perspective, and added more detailed and intuitive descriptions of QD in the appendix.
>
> >**Q5.** An ablation study highlighting the impact of various design choices would have added depth to the research.
>
> **A5.** We appreciate the suggestion. In this work, we explored two general design choices of QDHF in the aspect of training procedure, i.e., offline and online training. The ablation study focuses on the sample efficiency of QDHF by varying the number of human labels available, which further provides evidence of its scalability. We intentionally keep the implementation consistent across all three benchmark tasks, which follows a standard contrastive learning recipe.
>
> We would like to clarify that while there could be design choices such as using a higher dimension in the latent projection when modeling the diversity space for QDHF, we want to demonstrate in this work that with the standard design, QDHF can mirror the search capability of QD with manually crafted diversity metrics, and shows great performance consistently on all the benchmark tasks. We definitely agree that there could be other design choices that work well for each specific problem domain, which is related to the shape and complexity of the objective landscape. However, such studies do not completely align with the goal of this work, and therefore we look forward to exploring this topic in future work on specific domains.
>
>
> >**Q6.** Regarding the image generation experiment, I'm unclear about the source of randomness. Is the randomness exclusively attributed to the diffusion process? Or is there also an element of prompt modification? Additionally, could you elucidate how the cells are defined in image generation?
>
> **A6.** We appreciate the comment and added the following discussion to the paper for a better explanation of the LSI task. The source of randomness is the initial latent vector with dimensions (4, 64, 64), which is the input to the stable diffusion model. The default way to generate an image is to sample a latent vector randomly from uniform (0-1), and for the baseline best-of-n method, we generate 2000 images in such a way and select the images with the best CLIP score. We display the best 8 (or 16) images in the 2x4 (or 4x4) cells.
>
> For our method, the latent is optimized with QDHF such that each time we sample a latent from the archive and perform mutation by adding Gaussian noise (i.e., a general procedure in QD). The archive has a dimension of 20x20, so we will store at most 400 results. For qualitative visualization, we display the image in 4x4 cells by sampling from this 20x20 archive, i.e., each displayed image is the one with the best CLIP score in a 5x5 subset of the archive.

---

### Official Review · Reviewer_5CR4 · 2023-10-30

**Soundness:** 2 fair
**Presentation:** 2 fair
**Contribution:** 3 good
**Rating:** 3
**Confidence:** 2

**Summary:**

This paper investigates how to learn diversity metrics from human feedback, which is beneficial as typically diversity metrics must be manually specified. They propose a new algorithm, QDHF which uses human feedback to infer diversity metrics. QDHF can outperform the AURORA algorithm for diversity discovery.

**Strengths:**

Strength 1: Brings up interesting problem of how to learn diversity metrics from human feedback.

Strength 2: The proposed QDHF algorithm seems to be novel.

**Weaknesses:**

Weakness 1: The writing is not very clear. There are numerous typos and the importance / use cases of diversity metrics is not made clear in the introduction.

Weakness 2: This paper only uses one baseline algorithm (although several variants are considered) and this algorithm is unsupervised. The authors then claim that their algorithm can "outperform current QD" methods, but this is not at all a fair comparison. I think this difference in settings should be addressed more clearly.

Weakness 4: The qualitative evaluation of the LSI seems not very rigorous. In particular, I think that selecting images with the highest clip score in the left hand column may be incorrect, as selecting based on clip score alone could diminish the diversity. In addition, I think the authors should try to do a large scale/quantitative evaluation. Otherwise it is hard to tell if this scenario is cherry-picked. Moreover, I think that the authors should compare with some simple heuristics for increasing generation diversity, such as using different generation hyperparameters.

Weakness 3: This paper claims that they are expanding RLHF to consider the diversity of the generated responses/solutions, but the experiments do not show such results in typical RLHF environments and the paper does not compare their algorithm with standard RLHF.

**Questions:**

See weaknesses section.

---

> ### Author Response · Authors · 2023-11-22
> **Rebuttal from Authors of Submission6896 to Reviewer 5CR4 (1/2)**
>
> >**Q1.** The writing is not very clear. There are numerous typos and the importance / use cases of diversity metrics is not made clear in the introduction.
>
> **A1.** We appreciate the valuable comment and add the following discussion to the paper in the introduction. In optimization, diversity encourages exploration, which is essential for finding novel and effective solutions to complex problems. Without diversity, algorithms might converge prematurely to suboptimal solutions, resulting in getting stuck in local optima or producing only a limited set of responses (mode collapse). The diversity aspect is especially significant in quality diversity (QD) algorithms, where diversity metrics are explicitly utilized to encourage and maintain the variation of solutions during optimization.
>
> >**Q2.** This paper only uses one baseline algorithm (although several variants are considered) and this algorithm is unsupervised. The authors then claim that their algorithm can "outperform current QD" methods, but this is not at all a fair comparison. I think this difference in settings should be addressed more clearly.
>
> **A2.** We appreciate the reviewer’s suggestion. We agree with the reviewer that the claim "our method outperforms current QD" is not precise, since we are not comparing against classic QD algorithms. We add the following discussion in Sec. 2.1 to clarify the difference between QD methods. In general, a lot of QD algorithms, especially the MAP-Elites family, require manually crafted diversity metrics. AURORA is a recent method that extends QD with unsupervised learning of diversity metrics. Both AURORA and our method focus on applying QD in the scenario where there is no hand-crafted diversity metric, and AURORA is the only method in the literature we can compare against in this scenario.
>
> Regarding fair comparison, what the reviewer pointed out is correct. To our knowledge, our method is the first method that uses sparse human feedback in QD, as opposed to using more constructive human knowledge (e.g., crafting a diversity metric). While it is not possible to perform a comparison that is fair in terms of cost and labor, we design the experiments in a way that can validate our method to the largest extent within the scope of QD literature. We want to clarify that, first, we are not focusing on comparing against AURORA since our method additionally uses human feedback. AURORA just serves as a baseline here, and the results are to illustrate how much gain we can get from using QDHF.
>
> Secondly, what we would like to highlight is that, in terms of all solutions approached during training, QDHF-online closely matches the performance of QD with a ground truth metric, suggesting QDHF as a potent alternative for QD when manually designed metrics are not available for optimization, as shown in Sec. 4.3. These results establish a strong proof-of-concept of the benefit of QDHF, which aims to solve complex tasks where people find an unsupervised method is not sufficient, but designing a diversity metric is not feasible.
>
> In addition, we introduce two training strategies for QDHF where the QDHF-offline can be alternatively viewed as a stronger baseline for our method. QDHF-offline straightforwardly uses human feedback by learning a diversity metric first and then running QD. We highlight the implementation of QDHF-online, which actively learns the diversity during optimization as a deeper integration with QD, and shows much better performance.

---

> ### Author Response · Authors · 2023-11-22
> **Rebuttal from Authors of Submission6896 to Reviewer 5CR4 (2/2)**
>
> >**Q3.** The qualitative evaluation of the LSI seems not very rigorous. In particular, I think that selecting images with the highest clip score in the left hand column may be incorrect, as selecting based on clip score alone could diminish the diversity. In addition, I think the authors should try to do a large scale/quantitative evaluation. Otherwise it is hard to tell if this scenario is cherry-picked. Moreover, I think that the authors should compare with some simple heuristics for increasing generation diversity, such as using different generation hyperparameters.
>
> **A3.** We agree with the reviewer that optimizing the CLIP score alone could diminish the diversity, and that is a common problem in optimization, which motivates this work. In fact, the goal of the LSI task or more general image generation task is to obtain high-quality solutions, where best-of-n is a good baseline as a black-box optimization method (we assume that we do not have access to the model parameters in LSI).
>
> The presented result is an example prompt used in the release of Stable Diffusion, so we think it is a popular/representative prompt to demonstrate our method (not specifically picked according to the performance of our method). As suggested by the reviewer, we have run the LSI task on 5 other prompts in addition as proof of generality. We also include quantitative evaluation for the LSI task, where we add average CLIP scores for evaluating objective, and average pairwise similarity scores (using DreamSim) for evaluating diversity.
>
> In addition, we also follow the reviewer's suggestion and add another comparison algorithm for the LSI task. In the basic best-of-n approach, each time we sample a random latent with dimensions (4, 64, 64) from uniform (0-1), which is the default configuration for stable diffusion. To increase the diversity, we implement a simple heuristic that works as follows: Starting from the second image, for each step, we sample 100 latents randomly from uniform, and choose the latent that has the maximal l2 distance to the previous one. In this case, we increase the diversity of the latent, which is the source of randomness in generation. However, we find that this heuristic does not produce better diversity, which indicates that the stable diffusion model itself has some inductive bias in its generation, and generating diverse images is actually a challenging optimization problem on the latent space, which is essentially what the LSI task is aiming for. We show that QDHF outperforms both baseline methods in terms of diversity through quantitative and qualitative evaluation, and there is no significant difference in quality. The additional results are summarized in the table below and Table 3 in Sec. 4.3.
>
> | Method      | CLIP Score | Mean PD | Std. PD  |
> |-------------|------------|---------|---------|
> | Baseline    | 68.85      | 0.420   | 0.106   |
> | Baseline+   | 68.90      | 0.419   | 0.105   |
> | QDHF-online | **69.08**  | **0.527** | **0.151** |
>
> These results indicate that QDHF is able to generate more diverse solutions while maintaining the high quality of solutions.
>
> >**Q4.**  This paper claims that they are expanding RLHF to consider the diversity of the generated responses/solutions, but the experiments do not show such results in typical RLHF environments and the paper does not compare their algorithm with standard RLHF.
>
> **A4.** We appreciate the valuable comment and add the following discussion to the introduction and conclusion. This paper aims to broaden the scope of learning from human feedback (not RLHF specifically) to include optimizing for interesting diversity among responses, which is of practical importance for many creative applications. We are not comparing directly against RLHF. Instead, our method could be combined with RLHF for complex optimization tasks, where RLHF can be used as the objective for quality. The main reason we do not test in an RLHF domain is that RLHF is a relatively new paradigm, and there is no prior work that applies QD in such domains since it is hard to manually craft diversity metrics in those complex tasks. This work aims to provide a proof of concept that QDHF could mirror the capability of QD by evaluating standard QD tasks, and we look forward to applying our method in RLHF domains specifically in follow-up work.

---

### Official Review · Reviewer_XG8W · 2023-10-31

**Soundness:** 2 fair
**Presentation:** 2 fair
**Contribution:** 2 fair
**Rating:** 5
**Confidence:** 3

**Summary:**

This paper introduced QDHF, a Quality Diversity (QD) algorithm to learn diversity metrics from human feedback, instead of relying on manually defined metrics as in conventional QD algorithms. Contrastive learning and human judgement employed to align the learned metrics with human preference. Two training strategies are devised. Experiments are done on various AI tasks, such as Robotics, Reinforcement Learning (RL), and Computer Vision (CV).

**Strengths:**

1. The notion of integrating quality diversity and human feedback is both innovative and reasonable, and has been demonstrated to be efficacious.
2. The experiment was conducted in three disparate fields, which further substantiated QDHF is a ubiquitous approach for quality diversity.
3. In the context of latent space illumination, the author employs stable diffusion in conjunction with CLIP, instead of StyleGAN, which is a more sophisticated and functional approach.”

**Weaknesses:**

1. The authors mentioned several quality diversity (QD) method in Section 2.1, but the sole method of comparison is AURORA and the extension. It seems that the comparison is insufficient for evaluation and comparison.
2. Examples for latent space illumination (LSI) are somewhat partial. Some of the images contain unreasonable facts (such as a big brown helmet above the astronaut). More analysis is required.
3. Quality diversity is an important topic in dialog generation and image captioning. Yet the balance between diversity and rationality/accuracy is required. It seems that if the diversity is added, the rationality of generated results would be hampered. Therefore it would be better to evaluate both diversity and rationality at the same time.
4. Adding user study would be better for human-related tasks.

**Questions:**

Please check the weakness part.

---

> ### Author Response · Authors · 2023-11-22
> **Rebuttal from Authors of Submission6896 to Reviewer XG8W (1/2)**
>
> >**Q1.** The authors mentioned several quality diversity (QD) method in Section 2.1, but the sole method of comparison is AURORA and the extension. It seems that the comparison is insufficient for evaluation and comparison.
>
> **A1.** We appreciate the comment and apologize for the confusion. We have added the following discussion in Sec. 2.1 to clarify the difference between QD methods. In general, a lot of QD algorithms, especially the MAP-Elites family, require manually crafted diversity metrics. AURORA is a recent method that extends QD with unsupervised learning of diversity metrics, and it works on the top of existing QD algorithm. Similarly, our method learns diversity metrics in a semi-supervised way and also works with any existing QD algorithms. Both AURORA and our method focus on applying QD in the scenario where there is no hand-crafted diversity metric, and AURORA is the only method in the literature we can compare against.
>
> We also want to clarify that we are not accentuating the fact that our method outperforms AURORA since our method additionally uses human feedback, so AURORA just serves as a baseline here. What we would like to emphasize is that, in terms of all solutions approached during training, QDHF-online closely matches the performance of QD with a ground truth metric, suggesting QDHF as a potent alternative for QD when manually designed metrics are not available for optimization, as shown in Sec. 4.3.
>
>
> >**Q2.** Examples for latent space illumination (LSI) are somewhat partial. Some of the images contain unreasonable facts (such as a big brown helmet above the astronaut). More analysis is required.
>
> **A2.** We appreciate the reviewer’s valuable comment and we can confirm that we observed a few artifacts in the LSI results. We suspect that part of the reason may be the CLIP model served as the optimization target may be flawed. We have switched to a larger CLIP model (from ViT/B-16 to ViT/B-32) and re-generated the results, and we find the results are slightly improved.
>
> As suggested by the reviewer, we also conducted a more comprehensive analysis of the LSI experiment with 6 different prompts. We report the CLIP score and DreamSim pairwise distance (mean and std.) as quantitative metrics. We can see that QDHF has similar a CLIP score to both baseline methods, and a much
> higher mean and std. in the pairwise distance. These results indicate that QDHF is able to generate
> more diverse solutions while maintaining the high quality of solutions. These additional results are summarized in the table below and Table 3 in Sec. 4.3.
> | Method      | CLIP Score | Mean PD | Std. PD  |
> |-------------|------------|---------|---------|
> | Baseline    | 68.85      | 0.420   | 0.106   |
> | Baseline+   | 68.90      | 0.419   | 0.105   |
> | QDHF-online | **69.08**  | **0.527** | **0.151** |
>
>
> We also compare QDHF and Baseline qualitatively through a user study with 43 participants, where QDHF also outperforms baseline with a considerable margin on both the user preference ratio and user diversity perception ratio. The results are summarized in the table below and Table 3 in Sec. 4.3.
>
> | Method      | User Preference | User Perceived Diversity |
> |-------------|-----------------|---------------------|
> | Baseline    | 26.7%           | 8.91%               |
> | QDHF-online | **54.7%**       | **75.2%**           |
>
>
> More detailed results for the LSI tasks are included in Appendix C.

---

> ### Author Response · Authors · 2023-11-22
> **Rebuttal from Authors of Submission6896 to Reviewer XG8W (2/2)**
>
> >**Q3.** Quality diversity is an important topic in dialog generation and image captioning. Yet the balance between diversity and rationality/accuracy is required. It seems that if the diversity is added, the rationality of generated results would be hampered. Therefore it would be better to evaluate both diversity and rationality at the same time.
>
> **A3.** We appreciate the valuable suggestion from the reviewer and added the following discussion to the paper. As suggested by the reviewer, we provide additional quantitative results for a comprehensive evaluation of the LSI task regarding both diversity and rationality. We report the average CLIP score for assessing the objective for optimization, i.e., how well the generated images match the prompt. We also use DreamSim to calculate the pairwise distance between images in the solution set, and report its mean
> and standard deviation as measures of diversity. The mean pairwise distance indicates the average
> separation of images, and the standard deviation indicates the variability in terms of how the images
> are distributed. The results are presented above in **A2**, where we show that QDHF is able to generate
> more diverse solutions while maintaining the high quality of solutions.
>
>
> >**Q4.** Adding user study would be better for human-related tasks.
>
> **A4.** We appreciate the valuable suggestion from the reviewer and added a user study to our paper. We conducted a survey (n = 43) to assess their opinion on 1) if the QDHF results are more preferred, and 2) if the QDHF results are more diverse. We report
> the survey results as qualitative indicators for user experience. In addition, we also display samples
> of solutions for qualitative assessment. The results are also presented above in **A2**. QDHF also outperforms the baseline with a considerable margin on both the user preference ratio and user diversity perception ratio.

---

### Official Review · Reviewer_2snW · 2023-11-01

**Soundness:** 2 fair
**Presentation:** 2 fair
**Contribution:** 2 fair
**Rating:** 3
**Confidence:** 4

**Summary:**

Summary of the paper's contribution: This paper introduces Quality Diversity through Human Feedback (QDHF), a novel approach that combines human feedback with quality diversity (QD) algorithms to infer diversity metrics. QDHF aims to overcome limitations of reinforcement learning from human feedback (RLHF) and QD algorithms by leveraging human feedback for learning diversity metrics. The paper presents empirical results showing that QDHF outperforms existing QD methods in automatic diversity discovery and matches the search capabilities of QD with human-constructed metrics. In a latent space illumination task, QDHF significantly improves the diversity of images generated by a Diffusion model.

**Strengths:**

1. QDHF leverages human feedback to learn diversity metrics, expanding the applicability of QD algorithms. The proposed approach has the potential to improve exploration, personalization, and fairness in optimization for complex, open-ended tasks.
2. Empirical results demonstrate the effectiveness of QDHF in comparison to existing QD methods and its ability to enhance diversity in image generation tasks.
3. The paper is easy to understand.

**Weaknesses:**

1. The paper could provide more details on the scalability of QDHF, especially in the context of more challenging RL and open-ended learning tasks.
2. The evaluation metrics used in the paper are primarily quantitative, without fully leveraging the advantages of human feedback. For a method that relies on human feedback, more intuitive and persuasive evaluation criteria should include qualitative indicators such as user satisfaction and user experience.
3. The collection and processing of human feedback may be subject to subjective biases and inconsistencies. How to address these issues to improve the robustness and reliability of the method？
4. The method may be limited by the amount of human feedback data available. For large-scale and complex tasks, collecting sufficient feedback may be difficult and time-consuming.

**Questions:**

See weakness.

---

> ### Author Response · Authors · 2023-11-22
> **Rebuttal from Authors of Submission6896 to Reviewer 2snW (1/2)**
>
> > **Q1.** The paper could provide more details on the scalability of QDHF, especially in the context of more challenging RL and open-ended learning tasks.
>
> **A1.** We appreciate the valuable suggestion and have added the following discussion to the paper. As pointed out by the reviewer, scalability is definitely an essential factor to consider before applying QDHF in complex RL and open-ended tasks, which may be costly. In Sec. 4.4, we extend the previous analysis on sample efficiency to a discussion on the general scalability. First, in an ablation study on the robotic arm, we observe that the QD score is highly correlated with judgment prediction accuracy, which indicates that QDHF requires judgment to be correctly predicted using the projected latent. As a result, by evaluating the judgment prediction validation accuracy during the search process, we will have an idea of whether the current labels are enough for QDHF to perform well.
>
> Secondly, the judgment data for QDHF does not need to be entirely from humans. In the LSI experiments, we use a preference model (DreamSim) to source human feedback during training QDHF, and we show that a highly accurate preference model can also be used for QDHF as an alternative to online human labelers. Similar approaches have been widely used in RLHF-based LLM training [1,2], where a preference model is trained on a fixed amount of human feedback data, and used to predict rewards for fine-tuning the LLM.
>
> Combining the above two observations, we conclude that QDHF has good scalability towards more complex tasks because 1) we can anticipate its performance through online validation of judgment prediction accuracy, and 2) the samples used by QDHF can be sourced from a preference model trained on a fixed amount of human feedback data.
>
> [1] Stiennon, N., et al., (2020). Learning to summarize from human feedback.
>
> [2] Ouyang, L., et al., (2022). Training language models to follow instructions with human feedback.
>
>
> >**Q2.** The evaluation metrics used in the paper are primarily quantitative, without fully leveraging the advantages of human feedback. For a method that relies on human feedback, more intuitive and persuasive evaluation criteria should include qualitative indicators such as user satisfaction and user experience.
>
> **A2.** We appreciate the reviewer’s comment and added the following discussion to the paper. First, we would like to clarify that this paper aims to provide a proof-of-concept of the QDHF method with a focus on unbiased, quantitative evaluations, e.g., QD-scores with ground truth diversity metrics in robotic arm and maze navigation. For the LSI task where the ground truth diversity, we add CLIP scores for evaluating objective and mean pairwise similarity scores (using DreamSim) for evaluating diversity. The results show that QDHF demonstrates comparable quality to both baseline methods measured by CLIP score, and significantly better diversity measured by pariwise distance.
>
> However, we completely agree with the reviewer that for tasks such as text-to-image (which we believe could be a major use case of QDHF), more intuitive user-based qualitative indicators are more relevant. As suggested by the reviewer, we also added a user study to the LSI experiments. We ran LSI on 6 text prompts and a survey with a group of 43 participants. For each text prompt, we generate a 4x4 set of images for baseline (best-of-n) and QDHF, respectively, and ask the user: 1) which set of images they prefer, and 2) which set of images they think is more diverse. We summarize the results in the table below and Table 3 in Sec. 4.3. The qualitative indicators are ratios, and participants had the option to select 'hard to decide' in their responses.
>
> | Method      | User Preference | User Perceived Diversity |
> |-------------|-----------------|---------------------|
> | Baseline    | 26.7%           | 8.91%               |
> | QDHF-online | **54.7%**       | **75.2%**           |
>
> According to the user study, QDHF is considerably more preferable and generates more diverse responses as perceived by the users. We also include detailed user study results in Appendix C.

---

> ### Author Response · Authors · 2023-11-22
> **Rebuttal from Authors of Submission6896 to Reviewer 2snW (2/2)**
>
> >**Q3.** The collection and processing of human feedback may be subject to subjective biases and inconsistencies. How to address these issues to improve the robustness and reliability of the method？
>
> **A3.** We definitely agree with the review that human feedback may contain various kinds of noise, and that is part of the reason why we mainly use quantitative metrics in this work for evaluation. In the scope of using human feedback for QDHF, there are several ways to improve robustness. First, our implementation uses contrastive learning with a hinge loss, which tends to obtain the average preference given enough labels. As long as what the subjective biases or inconsistencies introduce is mainly variance in the preference data instead of population bias, stochastic optimization methods such as SGD are reasonably robust to this situation.
>
> Secondly, when working with large-scale problems such as LSI for image generation, instead of using human feedback directly, we use a preference model trained offline and source human feedback using the model when training QDHF. The preference model gives consistent and hopefully, unbiased preference predictions if the training data is at a large scale and the subjective biases from outliers can be averaged out.
>
> >**Q4.** The method may be limited by the amount of human feedback data available. For large-scale and complex tasks, collecting sufficient feedback may be difficult and time-consuming.
>
> **A4.** We agree with the reviewer that our method demands a reasonable size of human feedback data to learn a good latent representation of diversity, as described in Q1. However, QDHF accepts binary human preference from Two Alternative Forced Choice (2AFC), which is one of the easiest tasks for humans to label. In other words, the scalability of our method is at the same level as current RLHF applications such as text and image generation in terms of data collection, which means our method can be easily extended to many large-scale and complex tasks.
>
> Besides, as we explained in **A1**, the samples used by QDHF can be sourced from a preference model trained on a fixed amount of human feedback data, which means that we can collect feedback at a large scale prior to training, which can be parallelized and sourced in a timely manner.

---

### Author Response · Authors · 2023-11-22
**Summary of Changes in the Rebuttal Revision**

1. **Scalability of QDHF**: The paper expands on the scalability of QDHF in Section 4.4. It discusses how QDHF's performance can be anticipated through online validation of judgment prediction accuracy and how samples for QDHF can be sourced from a preference model trained on human feedback data.


6. **Quantitative Evaluation for LSI**: Additional analysis has been conducted for the LSI task with 6 different prompts. The paper now includes more comprehensive quantitative results, including CLIP scores and DreamSim pairwise distances. The results are presented in the table below:

   | Method      | CLIP Score | Mean PD | Std. PD  |
   |-------------|------------|---------|----------|
   | Baseline    | 68.85      | 0.420   | 0.106    |
   | Baseline+   | 68.90      | 0.419   | 0.105    |
   | QDHF-online | **69.08**  | **0.527** | **0.151** |

    These results indicate that QDHF not only maintains a high quality of solutions, as shown by similar CLIP scores to the baselines, but also achieves significantly greater diversity, evidenced by higher mean and standard deviation in pairwise distances.

2. **Qualitative Evaluation for LSI**: A user study was added to complement the evaluation of LSI. This study assesses user preferences and perceived diversity, showing that QDHF is more preferable and generates more diverse responses compared to the baseline. The results are summarized in the table:

   | Method      | User Preference | User Perceived Diversity |
   |-------------|-----------------|--------------------------|
   | Baseline    | 26.7%           | 8.91%                    |
   | QDHF-online | **54.7%**       | **75.2%**                |

    These results demonstrate a significant preference for QDHF over the baseline, both in terms of user preference and perceived diversity. In addition, detailed results of LSI including visualizations have been added to the Appendix for qualitative assessment.

4. **Human Feedback Data Requirements**: The authors discuss the feasibility of collecting human feedback data, noting that QDHF uses a simple and efficient method (2AFC) for gathering this data, making it scalable for large-scale tasks.

5. **Comparison with AURORA**: The paper now includes a clearer explanation of the choice of AURORA as a baseline for comparison. It clarifies that QDHF is not directly compared against AURORA but is presented as an alternative where human feedback can be utilized to significantly improve the performance of QD, in the same situation where predefined diversity metrics are not available.

7. **Balance Between Diversity and Rationality**: The authors have added quantitative results to evaluate both diversity and the objective quality of solutions generated by QDHF.

8. **Clarity and Accessibility of the Paper**: Efforts have been made to make the paper more accessible to readers not well-versed with quality diversity, including revising Section 2 and adding detailed descriptions in the appendix.

10. **Image Generation Experiment Details**: Clarifications have been added regarding the source of randomness in image generation and the definition of cells in the context of the stable diffusion model.

---

### Meta-Review · Area_Chair_Dt63 · 2023-12-12

**Metareview:**

Summary:

This paper presents an approach, QDHF, that combines human feedback with quality diversity (QD) algorithms to learn diversity metrics, eliminating the need for manually defined metrics in traditional QD methods. The alignment of learned metrics with human preferences is achieved through contrastive learning and human judgment. Empirical evaluations demonstrate that utilizing learned diversity metrics enhances results compared to standard diversity metrics across diverse tasks.

Strengths: Overall, the reviewers liked the idea of learning diversity metrics from human feedback and noted the consistent effectiveness of the proposed method across three distinct fields.

Weaknesses:

The primary weaknesses identified by the reviewers were centered around the evaluation, which did not convincingly showcase a clear advantage of the proposed QDHF over existing methods. During the rebuttal, the authors provided clarifications and additional results. Unfortunately, the reviewers did not actively engage in the discussion. Upon careful examination of the authors' rebuttal, the ACs share the view that the reviewers' concerns were not adequately addressed. Detailed comments are provided below:

Reviewer 2snW
- The reviewer expressed concerns regarding the scalability of QDHF, particularly in challenging RL and open-ended learning tasks. The examples provided in the authors' response are task-specific, leaving uncertainty about the generalizability of the conclusions to other tasks. Furthermore, the lack of direct up-scaling experiments raises questions about the validity of the scalability claims.

- The reviewer asked about qualitative evaluation. While the user study presented in the authors’ response is partially helpful, evaluating with only 6 text prompts makes the assessment less convincing.

Reviewer XG8W
- Regarding the analysis of LSI and rationality pointed out by the reviewer, the concerns remain, since 1) only 6 prompts seem inadequate, and 2) the gap between the baseline and QDHF seems very marginal, especially in the context of the CLIP score for the quality/rationality evaluation.

- There is a similar concern about the provided user study as discussed above.

Reviewer 5CR4
- There is a similar concern about the limited number of prompts regarding the LSI evaluation as discussed above.

- Regarding the suggestion to test against RLHF, the authors' response does not provide a solid justification, as the argument for not comparing in RLHF domains is that “it is hard to manually craft diversity metrics in those complex tasks."

Because of these unsolved concerns, the paper is not yet ready for acceptance. The ACs encourage the authors to address these issues and consider resubmission in the future.

**Justification For Why Not Higher Score:**

The current recommendation is based on the weaknesses summarized above: in particular, lack of convincing evaluation to demonstrate a clear advantage of the proposed work over existing methods.

**Justification For Why Not Lower Score:**

N/A

---

### Decision · Program_Chairs · 2024-01-16

Reject